# MRI-Cavernosography: A New Diagnostic Tool for Erectile Dysfunction Due to Venous Leakage: A Diagnostic Chance

**DOI:** 10.3390/diagnostics13132178

**Published:** 2023-06-26

**Authors:** Marco Di Serafino, Luigi Pucci, Francesca Iacobellis, Marco Fasbender Jacobitti, Roberto Ronza, Vittorio Sabatino, Luigi De Luca, Vincenzo Iossa, Nunzio Alberto Langella, Francesco Persico, Dario Grimaldi, Maria Laura Schillirò, Luca Lessoni, Maurizio Notorio, Maurizio Carrino, Luigia Romano

**Affiliations:** 1Department of General and Emergency Radiology, “Antonio Cardarelli” Hospital, 80131 Naples, Italy; iacobellisf@gmail.com (F.I.); roberto.ronza@hotmail.it (R.R.); vittorio.sabatino@gmail.com (V.S.); dariogrimaldi@me.com (D.G.); marialaura.schilliro@gmail.com (M.L.S.); luca.lessoni@aocardarelli.it (L.L.); maurizionotorio@libero.it (M.N.); luigia.romano1@gmail.com (L.R.); 2Department of Andrology, “Antonio Cardarelli” Hospital, 80131 Naples, Italy; luigi.pucci@hotmail.it (L.P.); m.fasbenderjacobitti@gmail.com (M.F.J.); luigideluca86@gmail.com (L.D.L.); vincenzoiossa@msn.com (V.I.); nunzio.lang@gmail.com (N.A.L.); francesco.persico90@gmail.com (F.P.); cris63@libero.it (M.C.)

**Keywords:** erectile dysfunction, venous leakage, color-doppler, cavernous CT, cavernous MRI

## Abstract

Erectile dysfunction caused by venous leakage is a vascular disease in which blood fails to accumulate in the corpora cavernosa due to the abrupt drainage of blood from the penis secondary to an abnormal venous network that affects 1 to 2% of men under 25 years old and about 10 to 20% over 60 years old, who do not raise a sufficient erection for penetrative sex. The study of the venous leak and its characterization in young patients with erectile dysfunction represent a diagnostic challenge, and imaging remains the best way to diagnose this condition. In the article, it is described the methods of execution and the diagnostic role of the cavernous MRI in the study of vasogenic erectile dysfunction from the venous leak, proposing it as a good alternative to the cavernous CT, considering the satisfactory results in terms of diagnostic interpretation, the absence of ionizing radiation, the higher soft tissue resolution of the imaging method and the lower administration of contrast agent.

## 1. Introduction

Erectile dysfunction (ED), previously named impotence, is defined as the “inability to achieve and/or maintain a satisfactory penis erection for satisfactory sexual performances” or “the constant or recurrent inability to achieve and/or maintain a sufficient penis erection for the sexual satisfaction” [1]. ED is a disease on continuous rise in the world because of the increase in expectation of life and chronic diseases such as diabetes, obesity, or hypertension, and it is estimated to involve 50% of men of age more than 40 years, having substantial effects on the quality of life [2,3].

Although some cases, particularly in younger men, may reflect psychological problems, in many cases, ED is caused by organic diseases, in particular, cardiovascular disease, diabetes mellitus, hyperlipidemia, and hypertension [4]; therefore, ED can act as a marker for medical conditions requiring treatment, representing a sign of a generalized vascular disease, as could also be suggested by the association with hemorrhoids [5,6,7]. 

Erection is a complex neuro-endocrine phenomenon, vascularly characterized by the dilatation of the afferent arteries to the corpora cavernosa (CC), with increased arterial flow rate, reduction or cessation of venous outflow and dilation of the sinusoidal spaces of the CC [2,3]. Penis erection is produced by an integration of physiological processes involving the central nervous systems, peripheral nervous, hormonal, and vascular systems that result in a relaxation of the tone of the fibrous smooth muscle cells of the arterial walls and of cavernous sinusoids, with a consequent increase in blood flow that fills and relaxes the gaps in the cavernous body, causing it to be mechanically closed and thus blocked in the venous outflow by means of veno-occlusive mechanism [6,7]. The albuginea, left to stretch at first, exhausts the maximum elasticity and becomes inextensible. Once the erection is achieved, the arterial flow tends to reduce, and rigidity is maintained by the venous discharge block. It then follows the detumescence, achieved by a reversal of the arterial-venous flows. This mechanism is characterized by vasoconstriction thanks to the activation of the sympathetic system, which returns to the state of tonic contraction of the stromal smooth muscle fibrocells, causing the squeeze of the cavernous tissue, with an increase in the venous return and the reduction of arterial inflow, such as to allow continuous circulation of blood at low pressure, which characterizes the state of flaccidity [8,9]. 

Based on the severity, ED can be classified according to the International Index of Erectile Function (IIEF) to stratify its entity [10,11]. Several physiopathologic mechanisms may sustain ED, as summarized in Table 1.

## 2. ED Due to Venous Leakage

ED caused by venous leakage is a vascular disease in which blood fails to accumulate in the CC due to the abrupt drainage of blood from the penis secondary to an abnormal venous network: blood reaches the penis through the arteries but escapes too quickly through abnormal veins; as a result, the pressure in the cavernous bodies cannot increase as much as “inflating a perforated balloon” [12].

Venous leaks are due to malformations of the penis’ veins or tunica albuginea. In other cases, venous leaks may occur due to the deterioration of the venous wall and the leakage of the endovascular valves [12,13].

The failure of venous occlusion, with early venous loss, is recognized as one of the most common causes of vasculogenic impotence in young patients [12]. Venous leaks have been present in some patients for as long as they can remember, and erections are difficult or even impossible since adolescence or soon after. This situation affects 1 to 2% of men under 25 years old and about 10 to 20% over 60 years old who do not have sufficient erections for penetrative sex [12].

The treatment essentially involves microsurgery, with the aid of vascular embolization techniques, which is effective; however, only in selected patients, typically healthy young men, with congenital or post-traumatic erectile dysfunction, resistant to medical therapy, allowing to delay the use of prosthetic implants that represents the definitive surgical treatment of choice for ED (Figure 1) [12,13,14,15].

## 3. Diagnostic Imaging Role in ED Due to Venous Leak

The study of the venous leak and its characterization in young patients with ED represents a diagnostic challenge. The microsurgical approach and vascular embolization techniques are assisted by an adequate anatomical study and precise evaluation of the escape sites [12,13,14,15]. As it is known, two distinct venous drainage systems are identifiable: a deep one, which drains the cavernous bodies and the glans through the deep dorsal vein directly into the retro-pubic plexus, and a superficial system, which drains the skin and adjacent tissues through the external pudendum veins directly into the large saphenous veins [12,13,14,15]. Imaging, therefore, remains the most appropriate recourse both for the selection of patients with possible erective deficit due to vascular etiology through the colour-Doppler and Doppler-pulsed study of the cavernous arteries after pharmacological stimulation (Figure 2) and to obtain anatomical details of the superficial and deep venous system of the penis [16,17,18].

X-ray cavernosography is the first conventional radiographic examination useful to define a venous vascular map of the penis and to identify abnormal venous leaks (Figure 3) [19]. 

However, this technique is extremely time-consuming and suffers from poor contrast and spatial resolutions with limited value in current clinical practice [12,15,20].

The limitations of conventional imaging have recently been overcome by tomographic study techniques through the use of cavernous-computed tomography (cav-CT), which provides high-resolution images of the venous system of the penis and perineum with an optimal representation of the venous anatomy, free from bone overlapping, with excellent diagnostic reliability regarding the presence of morphological alterations and potential venous leak pathways that allow for the planning of targeted surgical and non-surgical intervention strategies for eligible patients (Figure 4) [12,18,20].

Concerning the role of magnetic resonance imaging (MRI), while it has undoubted advantages over all other methods for the high contrast resolution in the study of the soft tissues constituting the penis [21], on the other hand, no extensive case studies are currently available on the validation of this imaging method such as an alternative to cav-CT in the study of venous leak ED, although with the indisputable benefits in terms of radiation protection and intrinsic contrast resolution. 

In the current technical note, we describe the methods of execution and the diagnostic role of the cavernous MRI (cav-MRI) in the study of vasogenic ED from a venous leak.

## 4. Methods and Results

We report our preliminary experience with the cav-MRI. The cav-MRI has been introduced in our Institute as a diagnostic indication in patients with suspected venous leak ED since 2019, an alternative to cav-CT, with the purpose of limiting patients’ radiation exposure and obtaining a better delineation of the anatomical structures. Until now, cav-MRI was performed on 35 patients, and in all of them, it demonstrated satisfactory results in terms of diagnostic interpretation, adopting the presented protocol.

### 4.1. Patient Selection

The questionnaire IIEF-6, the medical history, the objective examination, the routine blood tests, hormonal tests, and dynamic penile Doppler ultrasound were evaluated beforehand in all patients: patients with erectile dysfunction secondary to psychological disturbances, hormonal imbalances, or medication-related causes are excluded. In addition, patients with inadequate arterial inflow or mixed pathological changes on dynamic penile Doppler ultrasound examination, or with a known allergy to gadolinium chelate contrast agent, with claustrophobia, or with implantable devices incompatible with the magnetic field were not eligible. Eligible patients were those presenting with unsatisfactory erection, poor response to medical therapy, and increased diastolic flow in the cavernous artery (>10 cm/s) on dynamic penile Doppler ultrasound during an erection lasting longer than 10 min after intracavernosal injection of prostaglandins (PGE1).

### 4.2. Preparation and Implementation of the cav-MRI Protocol

Double informed written consent was obtained for both the intra-cavernous injection of gadolinium-based contrast agent and for the prior intra-cavernous administration of PGE1. The patients were placed in supine decubitus on the MRI bed. After local sterilization, 10 mg of PGE1 (Alprostadil, Caverjet, Pfizer, New York, NY, USA) were intra-cavernously administered, and a 21 G non-magnetic needle cannula was inserted into the dorsal side of a cavernous body at the third distal of the shaft. A safety venous needle cannula was in any way inserted into the ante-cubital forearm vein for any eventuality of adverse effects. The degree of erection was evaluated using the erection hardness score (EHS) that ranges from 1 (minimum) to 4 (maximum) [22]. In case of a non-erectile response, the protocol was repeated with a new administration of PGE1 with a maximum of 20 mg of PGE1. Obtained a satisfactory erection (EHS = 3 or 4), appropriate positioning of the patient was carried out by placing, in the supine decubitus, a towel of support under the thighs and the pelvis in order to elevate the penis and scrotum; the penis, in a state of erection, was stabilized on the anterior abdominal wall using a strip patch, with the precaution of not interfering with the proper placement of the cavernous access. A cotton ball can be maintained between the penis and the coil to avoid near-field artifacts. A multi-phase body surface coil (3–5 inches [7.62–12.70 cm]) was placed in correspondence with the penis region to maximize the signal-to-noise ratio. We used a high-field magnet (SIGNA^TM^, 1.5 T magnet, General Electric, Boston, MA, USA) with a dedicated study protocol, including a preliminary morphological phase through axial or multiplanar single-shot fast spin-echo (SSFSE) T2 weighted (T2W) sequences and a post-contrastographic, particularly venographic dynamic phase. The latter was obtained through the use of gradient echo (GRE) T1 weighted (T1W) 3D axial sequences with fat signal suppression before and after the intracavernous administration of the contrast agent (gadoteridolo; Prohance, Bracco S.p.A.; Milan, Italy) through the needle-cannula inserted in the CC at the base of the glans on the dorsal region, and connected to the automatic injector (MR PERION, Medrad-Bayer, software version: 1.10.206.06). One cubic centimeter (cc) of the paramagnetic contrast agent was then diluted in 20 cc of isotonic physiological solution and infused at a rate of 1.5 mL/s. Post-contrast images with tissue contrast enhancement, orthogonal multiplanar (MPR), and maximum intensity projections (MIP) reconstructions were obtained (Table 2 and Table 3).

Patients remained under observation until penis detumescence, ensuring that no procedural errors occurred, such as hematomas or contrast agent leakage.

### 4.3. Interpretation of cav-MRI Findings

All scans, including reconstructions, were carefully evaluated for a detailed morphological evaluation of the penis (Figure 5 and Figure 6) and the venous vascular anatomy, to identify any early or abnormal venous leak pathways (Figure 7, Figure 8 and Figure 9). 

## 5. Discussion

The study of the venous leak and its characterization in young patients with ED represent a diagnostic challenge; the imaging remains the best way to diagnose this condition. However, the patient should be preliminary carefully evaluated in the clinical history and clinically and laboratory assessed. In this regard, the IIEF-6 is a standardized questionnaire routinely used in patients with ED to stratify its entity. It considers five main issues: erectile function, orgasmic function, sexual desire, intercourse satisfaction, and overall satisfaction, and it was recently updated [23,24]. 

Patients clinically suspected of having ED, excluding those with ED secondary to psychological disturbances, hormonal imbalances, or medication-related causes, can then be sent to diagnostic imaging studies.

B-mode ultrasound and pulsed Doppler performed before and after drug-induced erection are the most widely used methods before resorting to invasive instrumental investigations. As is well known, the speed and morphology of the flow-metric curve detected at pulsed Doppler varies at different times during the erective process, in close dependence to the endocavernosal pressure, therefore observing physiologically high resistance flow spectra with inversion of the diastolic component during complete erection (Figure 10).

Otherwise, in the presence of increased venous outflow, a persistently high diastolic velocity is observed even at maximum erection obtained by drug induction: when the diastolic velocity is greater than 10 cm/s, venous etiology of ED should be suspected. An imaging contrastographic study for evaluation of the venous circulation must be carried out to confirm the diagnosis of ED by venous outflow (Figure 2) [18,25,26].

While cav-CT now represents the gold standard diagnostic reference for the diagnosis of ED from venous leakage, replacing traditional cavernosography due to its high spatial resolution providing a precise anatomical map of the penile, perineal, and pelvic venous districts as well as allowing an early detection of abnormal venous drainage vessels, superficial or deep, however, the radiation dose given to patients, often young, particularly on the gonads, cannot be neglected (3 mSv for each scan) with the related potential risks [15,27,28,29]. 

Consequently, it is really interesting to consider cav-MRI as a diagnostic alternative to cav-CT in eligible patients, with the possibility of similarly obtaining an anatomical map of the penile, perineal, and pelvic venous districts with MPR and MIP reconstructions, allowing the identification of sites of abnormal venous leakage, with marked diagnostic reliability as well as high contrast resolution for the anatomical structures of the penis that appears inherently superior to all other imaging methods [21,30]. 

Furthermore, the lower volumes of diluted contrast agent intra-cavernously administered (20 cc of cav-MRI vs. 20–60 cc of cav-CT) with a dilution ratio in physiological saline of the gadolinium-based contrast agent of 1:20 of cav-MRI vs. 1:2 of organo-iodine contrast agent used in cav-CT and lower injection rates (1.5 mL/s of cav-MRI as opposed to 2 mL/s of cav-CT), represent a further advantage of the method, which to date has not resulted in any allergic reactions to the contrast agent or post-injury compartmental syndromes. Maintaining a low, diluted dose of gadolinium-based contrast agent makes it possible to avoid the so-called pseudo-layering artifact in dynamic post-contrast sequences that occurs when there is a high concentration of the contrast agent and which can cause diagnostic interpretation errors in the study of CC (Figure 11).

However, the interpretation of cav-MRI results has limitations. Some of them are intrinsic to the method, such as artifacts from a ferromagnetic material, which may also depend on the use of non-magnetic needle cannulas (Figure 12).

Other limitations may instead be methodological. Among these, a non-negligible limitation that may invalidate the diagnostic deductions is represented by the much longer execution time of the investigation itself in comparison with cav-CT (30 min on average for the execution of a cav-MRI vs. 10–15 min on average for the execution of a cav-CT) with the risk of not adequately synchronizing the dynamic post-contrastographic study to the phase of maximum erection of the patient and therefore delaying the study analysis with respect to the erection obtained by the patient; if on an initial analysis of the SSFSE T2W sequences, no significant anatomical abnormalities are observed, in patients who are particularly anxious, it is possible to proceed directly to the dynamic post-contrastographic study in order not to prolong the time taken to perform the morphological study and therefore avoid “losing” the dynamic study and its diagnostic value during maximum erection.

Furthermore, the invasiveness of the cannulation of the CC has always been a limitation for erections, both because of the minimal loss of blood from the CC during erection and because of psychological factors such as anxiety or fear of experiencing pain during the cannulation of the CC. These aspects have, in some cases, led to a reduced erection or reduced duration of the erection, thereby affecting the diagnostic results; however, the latter is a limitation that applies to all methods of studying the CC with a contrast agent.

To the best of our knowledge, there are only two studies that test the role of MRI in venous leakage ED diagnosis, both through an intracavernous approach, such as our protocol [31], and through a traditional angiographic MRI study, the latter with contrast agent injection through the antecubital vein [30]. In the first reported study, the imaging protocol was not specified in detail, so limiting the reproducibility, and it is adopted a higher volume of intravenous contrast agent, whereas the second one is, in our opinion, limited by the acquisition protocol, since the CCs are not directly contrasted, the origin of the venous leak from the CCs themselves could be doubtful.

Furthermore, there are no structured scientific studies comparing the described diagnostic methods on the same patient in order to reinforce the current literature on the use of cav-MRI versus cav-CT in the diagnosis of ED from venous leakage. This is for various reasons, both for the recruitment of cases, i.e., patients undergoing cav-CT and subsequently cav-MRI and/or vice versa, but also for the comparison with controls, i.e., with the population of healthy patients who would deontologically not be eligible for the procedure and would clearly not be predisposed to it in the absence of a strong clinical motivation. However, the high quality of images obtained with cav-MRI with the interpretation of anatomical data and possible pathology, allow a positive balance between the diagnostic advantages obtained by the method and the intrinsic procedural limitations to cav-CT (Table 4).

A concretely achievable goal, however, is not the comparison between different imaging methods but consists in the evaluation of diagnostic results obtained with cav-MRI in comparison with the surgical findings on patients with ED from venous leakage unresponsive to medical treatment who have undergone surgery to ligate the deep dorsal vein of the penis or sclerotize it or a combination of them; patients currently undergoing follow-up to rule out any short- or long-term recurrences.

All the foregoing motivates our team to pursue this study protocol in keeping with the benefits and the high diagnostic potential of cav-MRI in order also to obtain statistically significant correlation data among imaging diagnosis, operative findings, and long-term prognosis.

## 6. Conclusions

Cav-MRI represents a promising imaging modality without ionizing radiation, useful for obtaining an accurate morphological and preoperative assessment of vasculogenic ED due to venous leakage.

The value of cav-MRI is crucial in selecting patients with ED due to venous leakage as these patients may benefit from surgical treatment, and the right identification of each individual venous leak pathway improves surgical outcomes with hopefully subsequent long-term good results and high patient satisfaction.

Patients who would benefit from surgical treatment are young patients with normal penile arterial systems and no risk factors.

Although extensive case studies are necessary to validate the method, we believe that the introduction of cav-MRI can improve the diagnosis of venous leaks with its detailed morphological and venous vascular anatomy information and serves as an alternative second-level imaging to cav-CT.

## Figures and Tables

**Figure 1 diagnostics-13-02178-f001:**
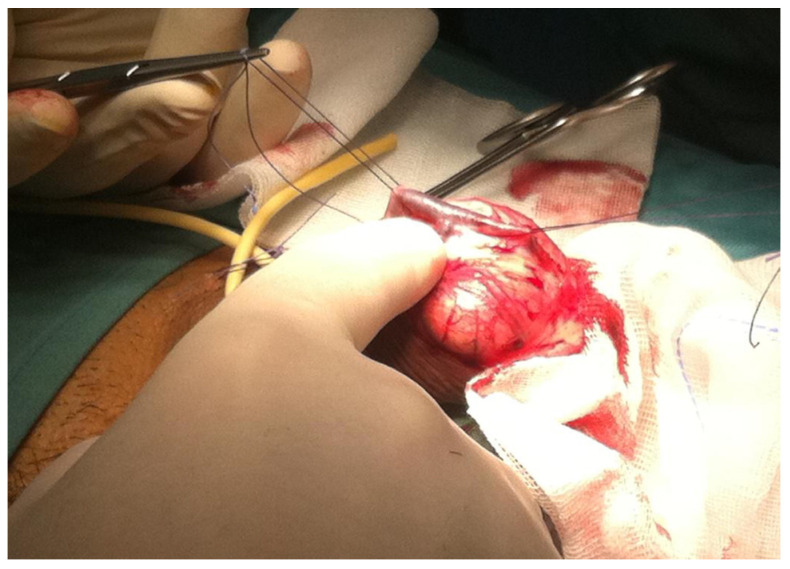
Surgical deep dorsal penile vein ligation.

**Figure 2 diagnostics-13-02178-f002:**
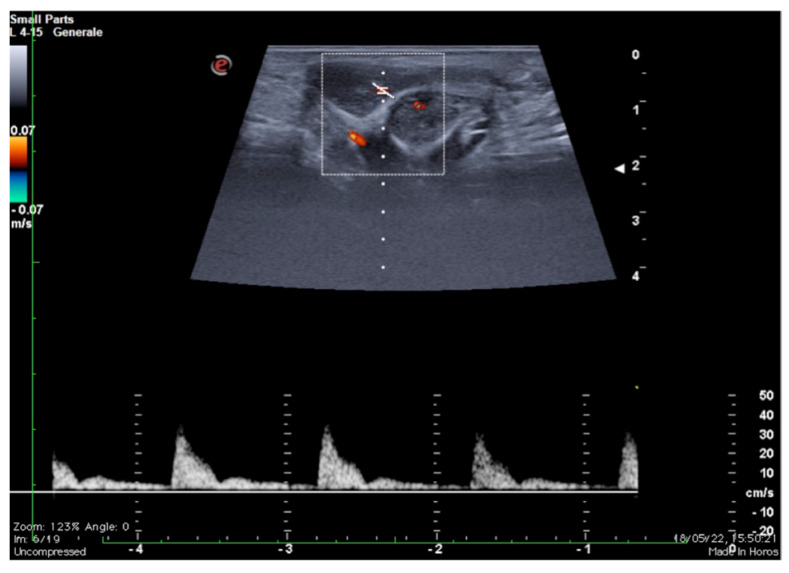
Functional impotence (erection duration) in a 39-year-old patient. An EHS 3 erection is observed 10 min after the injection of 10 mg of PGE1, characterized by a reduction in diastolic component but no reversal of tracing. Findings were suggestive of insufficient erection due to a venous leak. PGE1—prostaglandins; EHS—erection hardness score.

**Figure 3 diagnostics-13-02178-f003:**
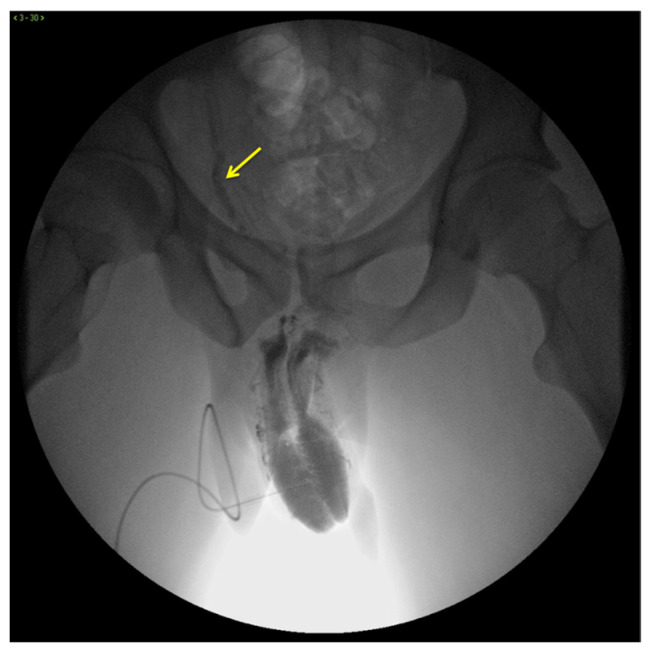
Cavernosography. Frontal view. Partial EHS 3 erection with fast deep venous outflow with right dominance (arrow). EHS–erection hardness score.

**Figure 4 diagnostics-13-02178-f004:**
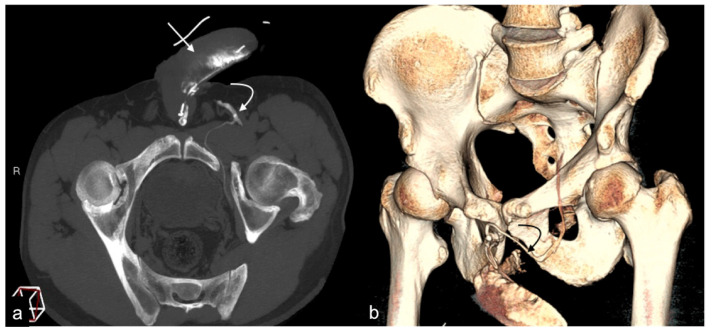
Cav-CT. MIP reconstruction on oblique axial plane (**a**) and VR (**b**). Partial erection with rapid outflow from the cavernous bodies ((**a**), straight arrow) related to significant leakage from pericavernous collaterals that drain into the hypogastric and left common femoral vein ((**a**,**b**), curved arrow). MIP—maximum intensity projection; VR—volume rendering.

**Figure 5 diagnostics-13-02178-f005:**
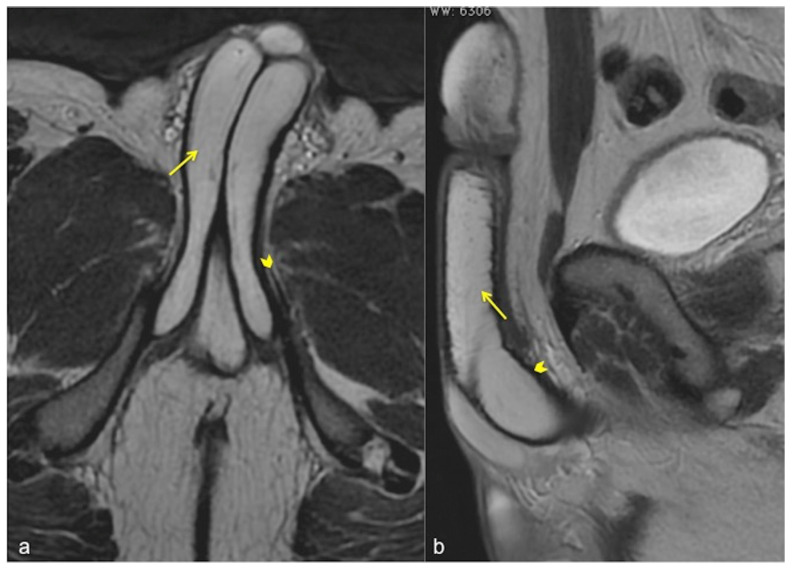
Normal anatomy of the penis. Axial (**a**) and sagittal (**b**) SSFSE T2W MRI images showing high signal intensity of the cavernous bodies and of the corpus spongiosum ((**a**,**b**), straight arrows), while the adjacent tunica albuginea has low signal intensity (arrowheads).

**Figure 6 diagnostics-13-02178-f006:**
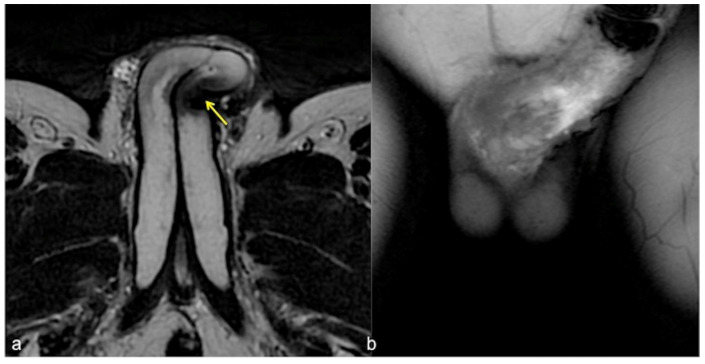
Cav-MRI. SSFSE T2W axial acquisition (**a**) showing a thickened plaque of reduced signal intensity in the left cavernous body ((**a**), arrow); GRE T1W 3D FS coronal acquisition obtained after contrast agent administration in the cavernous bodies (**b**) documents a defect in the contrast agent distribution due to fibrotic distortion of the cavernous bodies.

**Figure 7 diagnostics-13-02178-f007:**
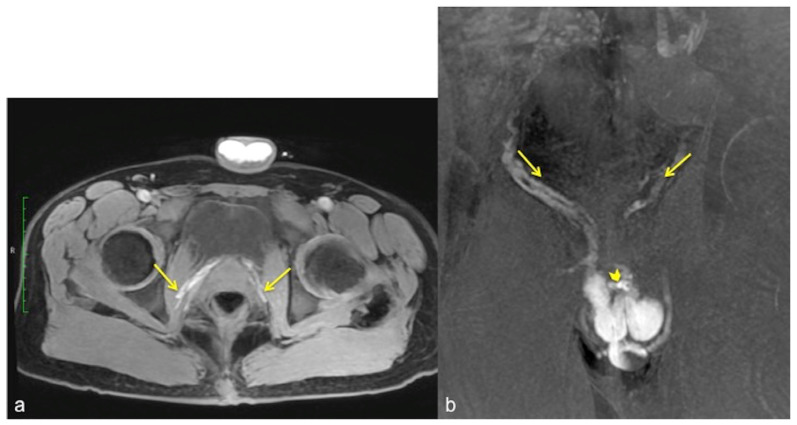
Cav-MRI. The axial (**a**) and coronal (**b**) GRE T1W 3D FS images after contrast agent injection, displayed with MIP reconstruction, show an early opacification of the crural veins ((**a**), arrows) and of the obturator veins ((**b**), arrows), supported by venous leakage mediated by the deep dorsal vein of the penis ((**b**), arrowhead).

**Figure 8 diagnostics-13-02178-f008:**
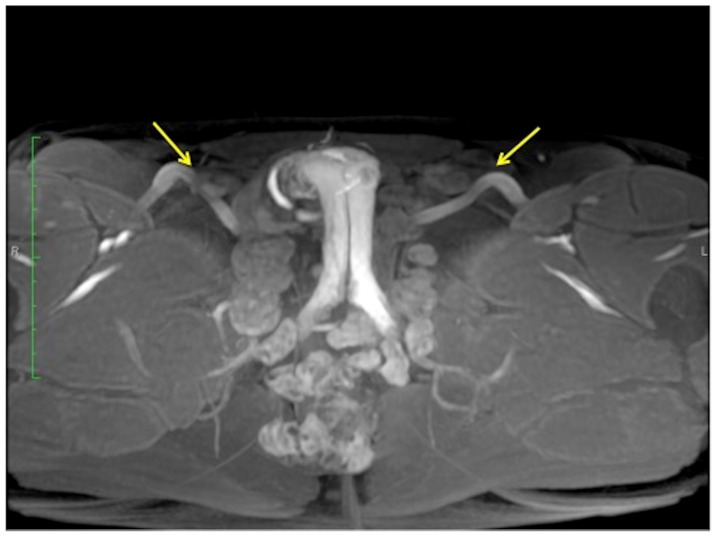
Cav-MRI. GRE T1W 3D FS axial image displayed with MIP reconstruction shows an ineffective erection supported by superficial venous leakage mediated by the external pudendal veins (arrows).

**Figure 9 diagnostics-13-02178-f009:**
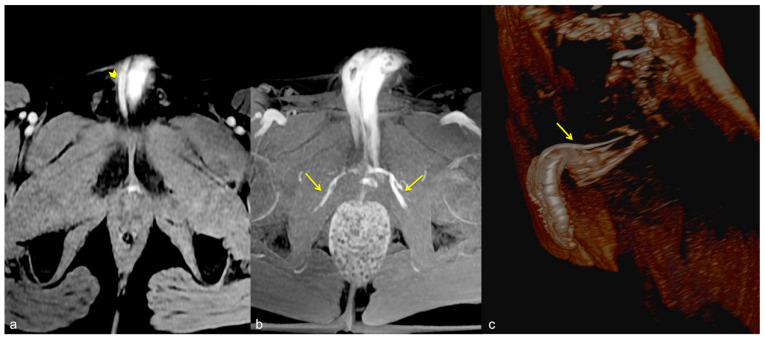
Cav-MRI. The GRE T1W 3D FS axial images displayed with MIP reconstruction after 1 min (**a**) and 5 min (**b**) from the intracavernosal injection of the contrast agent show an ineffective penis erection sustained by venous leakage mediated by the deep dorsal vein of the penis ((**a**), arrowhead) that feeds the obturator veins ((**b**), arrows). The 3D Volume-Rendering (**c**) reconstructed 1 min after the administration of the contrast agent documents the rapid penis detumescence with a dominant deep dorsal venous collector (arrow).

**Figure 10 diagnostics-13-02178-f010:**
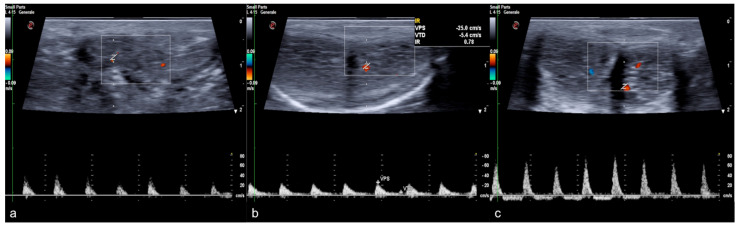
Color and pulse Doppler penile Ultrasound performed in different phases of erection. During the flaccid state (**a**), the waveform shows a monophasic pattern with minimal diastolic flow; during the tumescence (**b**), the waveform shows a progressive decrease in diastolic flow; during the full erection (**c**), the waveform shows a diastolic flow reversal.

**Figure 11 diagnostics-13-02178-f011:**
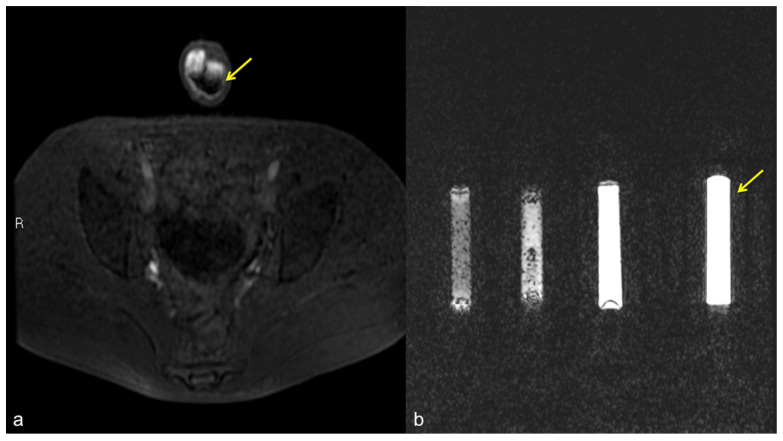
Pseudo-stratification effect of the contrast agent. Cav-RM GRET1W 3D fat sat axial sequence after intra-cavernous administration of contrast agent (**a**): pseudo-layering artifact of the contrast agent in the cavernous bodies (arrow). Saline dilution of the contrast agent in a test tube (**b**): from the left side of the figure, note the progressive optimization of the contrast agent signal from 5 cc, 3 cc, 2 cc, to the optimal ratio of 1 cc of contrast agent in a total volume of 20 cc of solution (contrast agent + saline) (arrow).

**Figure 12 diagnostics-13-02178-f012:**
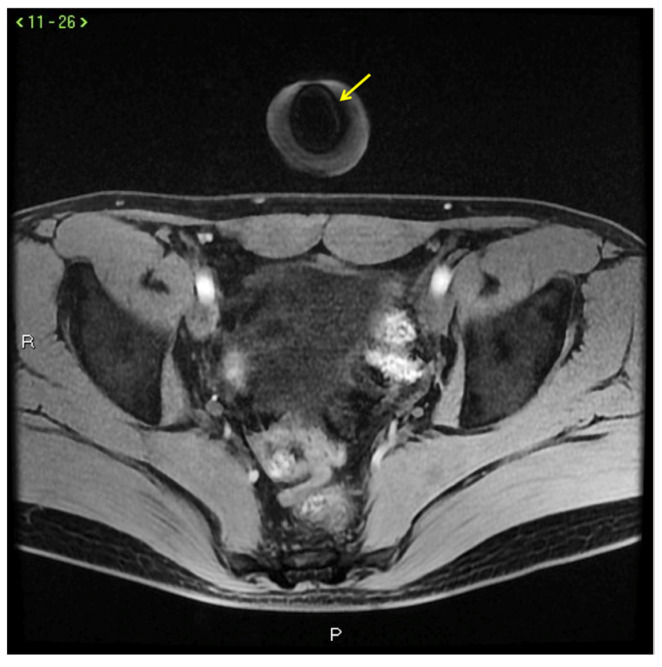
Signal voids (arrow) in CC caused by metallic needle cannula insertion.

**Table 1 diagnostics-13-02178-t001:** Different types of ED and their pathophysiologic mechanisms. ED: erectile disfunction.

Types of ED	Pathophysiologic Mechanisms
Psychogenic ED	The psychogenic component, consequent to various forms of neurosis, clearly prevails in the absence of organic damage.
Situational or relational ED	Form typically linked to a particular situation or specific partner and not present in other conditions.
Functional ED	When it is not possible to recognize any clear neurotic or psychotic pathology, but only a specific somatic “habitus” dominated by general symptomatology attributable to alpha-adrenergic hypertonia: sweating of the hands, tachycardia, etc.
Neurogenic ED	Secondary to diseases of the central or peripheral nervous system
Arterial or venous vascular ED	From arterial deficiency or veno-occlusive dysfunction
Iatrogenic ED	A side effect of pharmacological therapy or a sequel of a surgical intervention.
Endocrinology ED	Due to a deficiency of male sexual hormones (hypogonadism) related to an increase in prolactin (hypophysial hyperplasia/adenoma) or significant alteration in thyroid metabolism.
Diabetic ED	Due to vasculopathic or neuropathic etiopathogenesis diabetes-related.
ED related to chronic systemic diseases	Related to chronic kidney, liver, or heart diseases.
ED from congenital or acquired penile malformations	Phimosis, curvature, Induratio Penis Plastica.
ED from abuse of voluptous substances	Alcohol or drugs.

**Table 2 diagnostics-13-02178-t002:** Study protocol of cav-MRI.

Type of Acquisition *	Type of Sequence *	Results
Morphological acquisition	SSFSE T2 axial sequence with 3 mm slice thickness	Anatomical and morphological sequence
Optional supplemental morphological acquisitions depending on the pathology	Coronal and sagittal SSFSE T2W sequences with 3 mm slice thickness	Anatomical and morphological sequence (optional)
	SSFSE T2W fat sat axial sequence with 3 mm slice thickness	Anatomical and morphological sequence (optional)
Dynamic acquisitions for venographic study	Axial 3D GRE T1W fat sat with 2 mm slice thickness, prior to intracavernous administration of contrast agent	Anatomical and morphological sequence
	Axial 3D GRE T1W sequence with fat suppression with 2 mm slice thickness, after intracavernous administration of contrast agent	Dynamic sequences with MPR and MIP reconstructions,

* MRI sequences are acquired during erection and with planes orthogonal to the penis.

**Table 3 diagnostics-13-02178-t003:** Technical parameters.

Sequence	Phase Time	Plane	FOV	Matrix	Slice Thickness/Gap	TR (ms)	TE (ms)	Flip Angle	Fat Saturation
SSFSE survey	Pre-contrast	3 plane	39	256 × 158	7 mm/0 mm	6000	1–1.6	90	No
SSFSE T2W *	Pre-contrast	Axial	25	320 × 320	Mar-00	6000–8000	146	160	No
		Coronal	27	256 × 256	Mar-00	6000–8000	146	160	No
		Sagittal	25	320 × 320	Mar-00	6000–8000	108	160	No
SSFSE T2W FAT SAT	Pre-contrast	Axial	25	320 × 320	Mar-00	6000–8000	146	160	Yes
GRE T1W 3D FAT SAT	Pre-contrast	Axial	25	320 × 320	3	AUTO TR	MIN FULL	/	Yes
GRE T1W 3D FAT SAT	Post-contrast sequential dynamic 50 acquisition consecutively acquired starting with the injection of contrast agent for 5 min.	Axial	25	320 × 320	3	AUTO TR	MIN FULL	/	Yes

SSFSE: single shot fast spin echo; W: weighted; GRE: gradient echo; FOV: field of view; TR: time repetition; TE: time echo.

**Table 4 diagnostics-13-02178-t004:** Comparison between Cav-CT and Cav-MRI: advantage and limits.

Comparison between Cavernous CT and Cavernous MRI: Advantage and Limits		
Cavernous MRI		
Indications: Young patients Good compliance with MRI Suitable for magnetic field	Advantage	Good spacial resolution High contrast resolution Low volumes of contrast agent and reduced injection rate No ionizing radiations
	Limits	Magnetic fields (no conditional) Claustrophobia Quite long examination times
Cavernous CT		
Indications: Old patients Claustrophobic Weak compliance with MRI Not suitable for magnetic field	Advantage	Elevated spacial resolution Good resolution in contrast Fast acquisition times Post-processing
	Limits	Ionizing radiations Bigger volumes of contrast agent to high injection rate

## Data Availability

This study did not report any data.

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
