# Peer review of "MRI-Cavernosography: A New Diagnostic Tool for Erectile Dysfunction Due to Venous Leakage: A Diagnostic Chance"

_diagnostics, 2023, doi:10.3390/diagnostics13132178_

Round 1

Reviewer 1 Report (Previous Reviewer 2)

Thank you for inviting me to review the manuscript entitled „MRI-Cavernosography: A New Diagnostic Tool For Erectile 2 Dysfunction Due To Venous Leakage. A Diagnostic Chance” submitted for publication in the journal of Diagnostics. I appreciate the presentation of MRI-Cavernosography as a diagnostic method. The paper has got excellent illustrative material which increases its educational value.

I have seen this work during previous rounds of reviews and can tell that the work has been improved. I am glad that the authors decided to change the type of the manuscript from protocol to technical note as this type reflects better the content of the paper.

The abstract presents adequate information and the length is suitable for this type of paper.

The introduction presents sufficient information and background of the disease. However, in lines 56-61, classification should be presented according to eg. IIEF-5 questionnaire. You have got precise definitions of the severity of ED. Such a presentation would look more scientifically sound.

Please kindly improve the formatting of tables. For this consult the journal template.

Please check the abbreviations. Also, abbreviations used in tables should be explained under each table.

Author Response

Thank you.

Your suggestions were followed to further revise the manuscript.

Best regards,

The Authors

Reviewer 2 Report (New Reviewer)

The manuscript is well drafted and it can be accepted in the present form.

Its goog

Author Response

Thank you.

Best regards,

The Authors

This manuscript is a resubmission of an earlier submission. The following is a list of the peer review reports and author responses from that submission.

Round 1

Reviewer 1 Report

1. Please followup the author guideline for Protocol structure. The article format deviated to formal style.

2. The draft English should be revised by native experts. Some sentences are difficult to understand .

3. Introduction part: Part one and part two : to much repeated description  .

4. How many study subjects was performed  Cav-MRI in your study. All the Cav-MRI demonstrated satisfactory results?

5.  We can't get the real benefits of Cav-MRI versus Cav-CT in the discussion. Please reinforce the odds of in the discussion.

Please revised English by experts.  

Author Response

We would like to thank the Reviewer for their useful suggestion, helpful to improve manuscript quality.

Comments and Suggestions for Authors

  1. Please followup the author guideline for Protocol structure. The article format deviated to formal style.

We agree, reviewing the article styles we considered more appropriate to change the format in “Technical note” and we followed that style.

  1. The draft English should be revised by native experts. Some sentences are difficult to understand.

The article was further reviewed in the English form.

  1. Introduction part: Part one and part two : to much repeated description.

The Introduction was reviewed and rephrased.

  1. How many study subjects was performed  Cav-MRI in your study. All the Cav-MRI demonstrated satisfactory results?

Till now were examined 35 patients, and all the studies demonstrated satisfactory results in terms of diagnostic interpretation. These data were added in the methods and results section as follow: Till now cav-MRI was performed in 35 patients, and in all of them it demonstrated satisfactory results in terms of diagnostic interpretation, adopting the presented protocol”.

  1. We can't get the real benefits of Cav-MRI versus Cav-CT in the discussion. Please reinforce the odds of in the discussion.

Main benefits of Cav-MRI consists in the absence of radiation dose, high contrast resolution for the anatomical structures of the penis that appears inherently superior to all other imaging methods and in the lower volumes of contrast agent intra-cavernously administered. All of these advantages are reported in the table and in the article discussion.

Reviewer 2 Report

Thank you for inviting me to review the manuscript entitled „MRI-Cavernosography: A New Diagnostic Tool For Erectile 2 Dysfunction Due To Venous Leakage. A Diagnostic Chance” submitted for publication in the journal of Diagnostics. The work is interesting and within the scope of the journal. The authors describe the use of various diagnostic modalities for making the diagnosis of venous leakages within the penile vasculature. They provide their pros and cons which is useful for physicians. The paper has got excellent illustrative material which increases its educational value. Having this background information, the authors present their preliminary experience with the cavernous MRI.

My major concern is related to the type of this paper. It is not clear if the authors present their experience or if they present a study protocol for an upcoming clinical trial. This has to be clarified with the use of appropriate terminology.

Here are my minor comments that you can consider to improve the manuscript.

Abstract should be improved. The sentence in lines 20-24 is too long and too complicated. It should be divided into 2 or even 3 sentences. The abstract lacks conclusions. Please add information on the pros and cons of this method to give a straightforward message why it is worth to be used. And what are its superior features over other diagnostic methods?

Table 1 needs to be revised. In my opinion titles of columns do not correspond with the content. What is the intention of the right column? It is a definition of the type/etiology mentioned on the left one? Also, the content is not consistent. And the title is “risk factor for ED”. It also should be rephrased. Actually, this table is not needed, as you are going to focus on only one type of ED.

The introduction is messy. Please sort out the flow. In the first paragraph, you discuss comorbidities (line 36). Then there is classification, and then, there is a comeback to chronic diseases that serve as risk factors.

Abbreviations should be defined at first mention and then used only as abbreviations. In line 68, you use CC, but the term corpora cavernosa was used in the introductions several times already. Please, check the rest of the abbreviations.

Line 91, there is a typo “tecniques".

Title of Figures 2 and 4. Please explain the abbreviations used there.

Line 126, no need to capitalise the name of diagnostic procedures.

Line 135, this abbreviation was defined already in line 116.

Line 141, can you please write more about the IIEF-6 questionnaire? Please also add a reference to where the questionnaire was developed and validated.

Line 134. I think the title of this chapter is misleading. This is not an original study, in which you examine a certain population of patients and present the results. I would change is to e.g. “patient pathway with cav-MRI” or “cav-MRI – a procedure to diagnose penile venous leakage”. The title can also be different but should correspond to what is presented in this section.

Another issue is using the term “study protocol” which somehow does not fit here. If you want to present a study protocol for an upcoming trial it should be presented with greater detail with the use of the SPIRIT checklist.

Also in this chapter, please keep consistent grammatical tense. If you write “must be used”, it is not clear if such a procedure is used in your institution or it is just your recommendation.

Can you present some statistics on how many cav-MRIs are performed in your institution? Is it a routine examination?

Te paper has to be edited and properly formatted. There are plenty of typos and grammar inconsistencies. Please check carefully if paragraphs begin with proper intendicatios and are ended with hard breaks. I advise proofreading made by a medical editor.

Author Response

We would like to thank the Reviewer for their useful suggestion, helpful to improve manuscript quality.

Comments and Suggestions for Authors

Thank you for inviting me to review the manuscript entitled „MRI-Cavernosography: A New Diagnostic Tool For Erectile 2 Dysfunction Due To Venous Leakage. A Diagnostic Chance” submitted for publication in the journal of Diagnostics. The work is interesting and within the scope of the journal. The authors describe the use of various diagnostic modalities for making the diagnosis of venous leakages within the penile vasculature. They provide their pros and cons which is useful for physicians. The paper has got excellent illustrative material which increases its educational value. Having this background information, the authors present their preliminary experience with the cavernous MRI.

Thank you for your opinion

My major concern is related to the type of this paper. It is not clear if the authors present their experience or if they present a study protocol for an upcoming clinical trial. This has to be clarified with the use of appropriate terminology.

We agree and so we changed the format style in Technical note, more appropriate for this type of article.

Here are my minor comments that you can consider to improve the manuscript.

Abstract should be improved. The sentence in lines 20-24 is too long and too complicated. It should be divided into 2 or even 3 sentences. The abstract lacks conclusions. Please add information on the pros and cons of this method to give a straightforward message why it is worth to be used. And what are its superior features over other diagnostic methods?

The abstract was improved following your suggestion.

Table 1 needs to be revised. In my opinion titles of columns do not correspond with the content. What is the intention of the right column? It is a definition of the type/etiology mentioned on the left one? Also, the content is not consistent. And the title is risk factor for ED”. It also should be rephrased. Actually, this table is not needed, as you are going to focus on only one type of ED.

The table was revised according with your suggestion, if you agree, we prefer to leave it for completeness.

The introduction is messy. Please sort out the flow. In the first paragraph, you discuss comorbidities (line 36). Then there is classification, and then, there is a comeback to chronic diseases that serve as risk factors.

The introduction was rephrased

Abbreviations should be defined at first mention and then used only as abbreviations. In line 68, you use CC, but the term corpora cavernosa was used in the introductions several times already. Please, check the rest of the abbreviations.

All the abbreviations were reviewed, thanks

Line 91, there is a typo tecniques”.

Corrected, thanks

Title of Figures 2 and 4. Please explain the abbreviations used there.

Ok

Line 126, no need to capitalise the name of diagnostic procedures.

Ok, corrected

Line 135, this abbreviation was defined already in line 116.

Ok

Line 141, can you please write more about the IIEF-6 questionnaire? Please also add a reference to where the questionnaire was developed and validated.

More informations were added in the discussion section, as well as the references related to the development and validation.

Line 134. I think the title of this chapter is misleading. This is not an original study, in which you examine a certain population of patients and present the results. I would change is to e.g. patient pathway with cav-MRI” or cav-MRI – a procedure to diagnose penile venous leakage”. The title can also be different but should correspond to what is presented in this section.

We changed the format style in Technical note and the related article structure was followed.

Another issue is using the term study protocol” which somehow does not fit here. If you want to present a study protocol for an upcoming trial it should be presented with greater detail with the use of the SPIRIT checklist.

We changed the format style in Technical note and the related article structure was followed.

Also in this chapter, please keep consistent grammatical tense. If you write must be used”, it is not clear if such a procedure is used in your institution or it is just your recommendation.

Corrected

Can you present some statistics on how many cav-MRIs are performed in your institution? Is it a routine examination?

These data were added in the methods and results section as follow: The cav-MRI has been introduced in our Institute as a diagnostic indication in patients with suspected venous leak ED from 2019, alternative to cav-CT, with the purpose to limit patients radiation exposure and to obtain a better delineation of the anatomical structures. Till now cav-MRI was performed in 35 patients, and in all of them it demonstrated satisfactory results in terms of diagnostic interpretation, adopting the presented protocol”.

Round 2

Reviewer 1 Report

  Indeed the manuscript improved after revision. However, some major and serious concerns remained: 

1.As your statement " Institutional Review Board Statement: Ethical review and approval were waived due to the nature of the manuscript. Informed Consent Statement: Patient consent was waived as due to the nature of the manuscript."

All the studies need registration and approval of institutional IRB. Why patients waive their  informed consent if  this a new diagnostic technique? 

2. It is obviously not  a new diagnostic technique. It had been reported":   . Magnentic resonance imaging as a potential tool for objective visualization of venous leakage in patients with veno-occlusive erectile dysfucntion." in  Int J Impot Res2008 Mar-Apr;20(2):192-8.   On the other hand, you did not analyze your 35 MRI-cavernosography clinical parameter to present as  a research article. It is a protocol ?  a  new technique ? or a research article ?

Author Response

1.As your statement " Institutional Review Board Statement: Ethical review and approval were waived due to the nature of the manuscript. Informed Consent Statement: Patient consent was waived as due to the nature of the manuscript."

All the studies need registration and approval of institutional IRB. Why patients waive their  informed consent if  this a new diagnostic technique? 

We are sorry for the mistake, actually, as written in the manuscript page 6, at the beginning of the paragraph 4.2 “Double informed written consent is obtained for both the intra-cavernous injection of gadolinium based contrast agent and for the prior intra-cavernous administration of PGE1”. 

The study of the corpora cavernosa is a consolidated investigation in our Institution since 1995, through X-ray cavernosography, which later became CT-cavernosography and, only recently, MRI-cavernosography. The procedure, particularly the injection of the contrast medium into the corpora cavernosa, has remained unchanged since 1995, but the imaging acquisition has changed and has become MRI through an imaging protocol already dedicated to the study of the male pelvis and prostate. So, we have unified the execution of intracavernous injection with the MRI study, both of the procedures already known in the literature and in our Institute and therefore not to be considered experimental but already individually applied. Furthermore, the injection of only 1 cc of gadolinium chelated contrast agent into the corpora cavernosa responds to its prescribed application of intravenous injection. So, institutional approval was considered unnecessary and the informed consent statement was corrected. About the patients images, all of them are anonymized.

2. It is obviously not  a new diagnostic technique. It had been reported":   . Magnentic resonance imaging as a potential tool for objective visualization of venous leakage in patients with veno-occlusive erectile dysfucntion." in  Int J Impot Res. 2008 Mar-Apr;20(2):192-8.   On the other hand, you did not analyze your 35 MRI-cavernosography clinical parameter to present as  a research article. It is a protocol ?  a  new technique ? or a research article ?

The article is a “Tecnhical note”, as specified in the title page, and has the aim to describe in detail the methods of execution and the diagnostic role of the cavernous-MRI in the study of vasogenic erectile dysfunction from venous leak proposing it as a good alternative to the cavernous CT. So, the  specific analysis of the clinical parameters of our patient cohort goes beyond the aim of the present study and it will certainly be the subject of a subsequent publication in a dedicated research article.

In the discussion section was reviewed the paragraph in which were already reported and discussed the two available studies regarding MRI-cavernosography: “To the best of our knowledge, there are only two studies that test the role of MRI in venous leakage ED diagnosis both through an intracavernous approach, such as our protocol [Kurbatov DG, Kuznetsky YY, Kitaev SV, Brusensky VA. Magnetic resonance imaging as a potential tool for objective visualization of venous leakage in patients with veno-occlusive erectile dysfunction. Int J Impot Res. 2008 Mar-Apr;20(2):192-8. doi: 10.1038/sj.ijir.3901607], and through a traditional angiographic MRI study, the latter with contrast agent injection trough the antecubital vein [Roudenko A, Wilcox Vanden Berg RN, Song C, Prince MR, Paduch DA, Margolis D. Utility of dynamic MRA in the evaluation of male erectile dysfunction. Abdom Radiol (NY). 2020 Jul;45(7):1990-2000. doi: 10.1007/s00261-019-02339-y. PMID: 31784778.]. In the first reported study, the imaging protocol was not specified in detail, so limiting the reproducibility, and it is adopted an higher volume of intravenous contrast agent, whereas the second one is, in our opinion, limited by the acquisition protocol as while waiting for the venous phase obtained after the injection of contrast agent in the systemic venous circulation, penile detumescence could occurs and moreover, since the CCs are not directly opacified, the origin of the venous leak from the CCs themselves could be doubtful.” 

Furthermore in the article Kurbatov DG, Kuznetsky YY, Kitaev SV, Brusensky VA. Magnetic resonance imaging as a potential tool for objective visualization of venous leakage in patients with veno-occlusive erectile dysfunction. Int J Impot Res. 2008 Mar-Apr;20(2):192-8. doi: 10.1038/sj.ijir.3901607 in all the images are reported patients details!

We are available for further clarifications.

Hoping that the manuscript can now be considered for publication.

Kind regards,

The Authors

Reviewer 2 Report

Thank you for your work. I can see that the manuscript has been revised extensively. In my opinion, it can be published.

Author Response

Thank you for your opinion.

Kind regards,

The Authors

Round 3

Reviewer 1 Report

It depends on the identity of editorial office on the IRB status and suitability on technique note section.